# A Survey of LLM-based Multi-agent Systems in Medicine

## Abstract

Large Language Model (LLM)-based multi-agent systems have shown great potential in supporting complex tasks in the medical domain, such as improving diagnostic accuracy and facilitating multidisciplinary collaboration. However, despite the advancement, there is a lack of structured frameworks to guide the design of these systems in medical problem-solving. In this paper, we conduct a comprehensive survey of existing medical multi-agent systems, and propose a medical-specific taxonomy along three key dimensions: team composition, medical knowledge augmentation, and agent interaction. We further outline several future research directions, such as incorporating human–AI collaboration to ensure that human expertise and multi-agent reasoning jointly address complex clinical tasks, designing and evaluating agent profiles, and developing self-evolving systems that adapt to evolving medical knowledge and rapidly changing clinical environments. In summary, our work provides a structured overview of medical multi-agent systems and highlights key opportunities to advance their research and practical deployment.

## 1 Introduction

Large Language Model (LLM)-based multi-agent systems have recently gained significant attention for their potential to support complex decision-making processes in the medical domain. By leveraging the complementary reasoning and collaboration of multiple agents, such systems aim to address the limitations of single-agent approaches, including hallucination, lack of domain specialization, and difficulties in handling multi-step reasoning. Recent studies have explored their use in diverse medical tasks, such as clinical diagnosis (Chen et al., 2025d; Wang et al., 2025e), clinical triage (Lu et al., 2024), and clinical trial design and optimization (Yue et al., 2024). These applications highlight the promise of multi-agent systems in improving reliability, interpretability, and scalability in healthcare-related AI solutions.

Despite these advances, designing effective multi-agent systems for medicine remains highly challenging. To address this, an increasing number of studies have proposed various approaches, such as optimizing agent role allocation and collaboration strategies (Kim et al., 2024; Xia et al., 2025), incorporating domain knowledge into agent communication (Wang et al., 2025e), and designing mechanisms to organize agents' discussion processes (Wang et al., 2025c). For example, MDAgent introduces a framework that dynamically determines whether LLM-based agents should work individually or collaboratively according to the complexity of medical tasks, mirroring real-world medical decision-making (Kim et al., 2024). As the number of these efforts continues to grow, they remain fragmented, underscoring the need for a systematic understanding to help researchers clearly grasp the current landscape of these approaches and highlight future opportunities for designing multi-agent systems in medicine.

Several survey papers have begun to examine the development of multi-agent systems. However, many are domain-agnostic and therefore overlook medical-specific characteristics, such as designing agents to reflect the clinical workflows and physician specializations (Li et al., 2024; Guo et al., 2024). Some only focus on investigating the use of multi-agent systems for hospital simulation like hospital operations (Yao & Yu, 2025; Guo et al., 2024). A few surveys do discuss medical applications, but they either emphasize usage scenarios (Alshehri et al., 2023) or do not highlight the unique aspects brought by multiple agents, such as their interactions (Wang et al., 2025d). Moreover,

existing general taxonomies—for example, interaction modes like centralized, decentralized, hierarchical, and shared message pool (Li et al., 2024)—are also insufficient for medical contexts, as they overlook domain-specific mechanisms in clinical tasks, such as multidisciplinary team (MDT)-style collaboration or stage-wise clinical task allocation. To our knowledge, no prior survey has systematically structured existing work from this design-oriented perspective, nor provided a medical-specific coding framework that better reflects how multi-agent systems can be designed and deployed in real-world medical problem-solving tasks.

To fill this gap, our survey takes a design-oriented perspective on medical multi-agent systems. Specifically, we target to answer three questions: (1) *how to compose a multi-agent team to address practical problems*, (2) *how to empower agents with medical knowledge*, and (3) *how agents interact with each other*. To address these questions, we collected 50 papers that specifically focus on using LLM-based multi-agent systems for real-world medical problem-solving. We first analyzed these works and derived a medical-specific taxonomy along three dimensions: team composition, knowledge enhancement, and interactions, as shown in the appendix ( Table 1). We further discuss broader challenges in current approaches and outline promising directions for the future development of medical multi-agent systems. These include incorporating human–AI collaboration to ensure that human expertise and multi-agent reasoning jointly address complex clinical tasks; enabling safe and transparent self-evolution in response to new medical knowledge and rapidly changing clinical environments; achieving deeper multimodal integration for richer cross-modal reasoning; designing and evaluating agent profiles that balance realism with adaptability; and expanding to more diverse clinical and public health scenarios. We hope that our work serves as a starting point for researchers and practitioners to better understand and design next-generation medical multi-agent systems.

## 2 TAXONOMY OF MEDICAL MULTI-AGENT SYSTEMS

To build a comprehensive paper corpus, we first conducted a keyword search in Google Scholar using the query: *(LLMs OR large language models) AND (multi-agent OR multiple agents) AND (medicine OR medical OR clinical OR diagnosis)* and collected 15 seed papers. Starting with these seed papers, we iteratively expanded the corpus by tracing references. This process continued until saturation, when no new relevant papers were found, resulting in a total of 64 papers. Next, two authors independently reviewed the abstracts and introductions to exclude works unrelated to medical problem-solving (e.g., general-purpose systems (Wang et al., 2025b), hospital simulations (Zhuang et al., 2025; Almansoori et al., 2025), or purely biological or genomic research (Fan et al., 2025)). This filtering yielded 50 papers for the subsequent analysis. We then analyzed these papers along three design dimensions of multi-agent systems: team composition, medical knowledge enhancement, and agent interaction. Each paper was independently coded by at least two authors. Note that, at this stage, we did not have unified the terms for all the coded perspectives. Then, all authors conducted the weekly meeting to discuss and iterate the coding results until consensus was reached. This process resulted in a final taxonomy that captures the key design patterns of medical multi-agent systems, as shown in the appendix (Table 1 and Figure 1). In the following, we present this taxonomy across three aspects: team composition, medical knowledge augmentation, and agent interaction.

### 2.1 TEAM COMPOSITION

Team composition determines how agent roles are configured to address medical tasks. We identify five approaches to configuring the roles of agents in existing medical multi-agent systems.

**Clinical task allocation**: Agent roles are defined according to specific clinical tasks (e.g., diagnosis, prognosis, treatment planning) based on task requirements. Clinical task allocation emphasizes the organization of agent roles around specific medical tasks. Clearly defined task boundaries provide a foundation for subsequent task-level performance monitoring and behavioral pattern analysis. A representative example is presented by Chen et al. (2025f), who developed a multi-agent system for managing inpatient pathways. In their framework, agents such as the Admission Agent, Diagnosis Agent, Treatment Agent, and Discharge Agent are each responsible for distinct clinical phases, collaboratively covering the full patient journey from admission to discharge. Clinical task allocation is also applicable to finer-grained task decomposition in system design. For instance, Iapascurta

et al. (2025) designed a three-agent system where agents were responsible for the overall condition assessment, antibiotic recommendation, and compliance checking against clinical guidelines.

**Specialization-oriented assignment**: Agents roles are aligned with distinct medical specialties (e.g., radiology, pathology, pharmacology). This mirrors the role structures in real-world hospitals, enhancing diagnostic accuracy and ensuring strong system scalability. For instance, Zhou et al. (2025b) introduce a system where a General Practitioner performs triage and refers patients to a team of expert agents, each handling domain-specific tasks such as imaging interpretation or information synthesis. A Director agent then coordinates the discussion and generates the final diagnostic report. Similarly, Xia et al. (2025) use a Triage Doctor to route cases to specialists, whose opinions are integrated by an Attending Physician. This structure effectively simulates multidisciplinary teams (MDTs), where specialists collaborate to improve care for complex cases. Li et al. (2025b) design an MDT-inspired agent team for Alzheimer's diagnosis, where agents such as the Primary Care Physician, Neurologist, Psychiatrist, and Geriatrician each focus on complementary aspects of patient assessment. Their findings are synthesized by an AD Specialist agent to produce a final risk evaluation. This design enables agent-level simulation of MDT collaboration and enhances diagnostic performance in complex cognitive disorders.

**Process-oriented allocation**: Agent roles are defined based on stages of the decision-making or task completion workflow, such as planning, analysis, refine, and final decision-making. Compared to clinical task allocation and specialization-oriented assignment, which emphasize real-world clinical practices and role-specific responsibilities, process-oriented allocation assigns conceptual task flows to agents. By leveraging abstract cognitive structures to restructure problem-solving pathways, it transcends the constraints of conventional clinical thinking and enables more innovative and systematic forms of intelligent collaboration. For example, in the "Generation—Verification—Reasoning" task flow proposed by Hong et al. (2024), the Generator agent generates preliminary diagnostic or treatment plans based on predefined argumentation templates; the Verifier agent challenges these plans by posing structured critical questions, prompting the system to generate rebuttals or alternative arguments; finally, the Reasoner agent synthesizes the discussion and arrives at an acceptable decision. Similarly, Xu et al. (2025) designed three corresponding agents based on the "Generation—Evaluation—Optimization" task flow, responsible for generating initial proposals, evaluating the proposals, and optimizing them. This approach can be used to ensure the reliability and safety of medical decisions incorporating roles for review, feedback, and refinement.

**Expertise-level assignment**: Agent roles reflect different expertise levels, such as junior vs. senior physician roles. For instance, Ke et al. (2024b) implemented a multi-tiered diagnostic review system in which Junior Resident I was responsible for the initial diagnosis, followed by Resident II who critically evaluated the diagnosis from a peer-review perspective, aiming to identify cognitive biases such as anchoring bias and confirmation bias. A Senior Physician played a supervisory role, identifying and correcting cognitive distortions, and offering guidance and decision-making support. Low et al. (2025b) proposed a risk-aware routing mechanism for the delegation of surgical error detection tasks across three professional tiers: the Resident-level, Attending-level, and Expert-level. Agents at the resident level employed checklist-based conservative reasoning; attending-level agents integrated structured evaluations with contextual interpretations; and expert-level agents incorporated multi-scale temporal pattern recognition to provide high-level insights. Expertise-level assignment simulates multi-tiered collaboration among physicians of varying seniority, effectively enhancing the depth of decision review and the correction of cognitive biases.

**Automatic assignment**: Agent roles are automatically defined or selected by algorithms or optimization strategies that generate or select the most suitable agents for a given medical task. Dynamically selecting optimal agents through algorithmic strategies significantly enhances the system's adaptability and generalization across diverse task scenarios. For instance, Zhou et al. (2025b) constructed a domain-specific expertise table for various LLMs, systematically quantifying each model's strengths across medical domains, which enables the system to recruit an optimal subset of agents with demonstrated proficiency in the relevant subject areas and query difficulties. Yang et al. (2025) proposed the Rotation Agent Collaboration (RAC) mechanism, in which a leading agent is dynamically selected based on the inferred intent of the question. This agent gathers information through polling from other agents and, after fusing the responses, designates the most suitable agent to make the final decision. Zhao et al. (2025a) adopted a strategy of dynamically selecting AssistAgents aligned with the medical domains relevant to each query, assigning each agent to retrieve and synthesize evidence within its area of expertise. For medical question answering, Wang et al. (2025c)

generated domain-specific agents based on the domains associated with both the question and the answer options. To address challenges in rare disease diagnosis and treatment, Chen et al. (2024) designed an Attending Physician Agent that selects the most relevant specialists from a predefined pool based on the patient's clinical profile and forms a MDT to reach diagnostic consensus.

## 2.2 MEDICAL KNOWLEDGE AUGMENTATION

Equipping agents with medical knowledge is essential to ensure reliable reasoning and improve task accuracy in clinical contexts. Existing approaches can be broadly grouped into two categories: *agent-intrinsic* methods, which enhance knowledge within the agent, and *externally-assisted* methods, which integrate external knowledge sources to agents but not modify the agent models.

### 2.2.1 AGENT-INTRINSIC

Agent-intrinsic methods enhance the medical knowledge embedded within the agents themselves. These approaches represent a spectrum of increasing specialization, ranging from lightweight prompt engineering to more intensive modifications of the model's underlying parameters. This progression allows for tailored enhancement of an agent's expertise, moving from broad role simulation to deep, task-specific knowledge integration.

**Role-play prompting**: Agents are guided by assigning them specific medical roles (e.g., cardiologist, nurse, patient) through carefully crafted prompts. By framing the task within a professional persona, the LLM is encouraged to adopt a communication style, reasoning process, and knowledge domain relevant to that role. This is a zero-shot, computationally efficient method to improve the quality and relevance of the agent's output without altering the base model. For instance, several frameworks simulate multi-disciplinary team (MDT) consultations via role-play (Liu et al., 2025; Ke et al., 2024a). A typical example is MedAgents (Tang et al., 2023), which introduces a framework for addressing domain-specific terminology and expert reasoning in the medical field using large language models. By enabling the model to "role-play" different medical experts, MedAgents facilitates multi-turn collaborative discussions to analyze and solve medical problems, thereby improving reasoning accuracy and interpretability.

**Pre-trained model utilization**: Agents leverage LLMs pre-trained on large-scale and curated medical datasets, instead of general-purpose LLMs. This approach overcomes the inherent knowledge limitations of generalist models by providing medical knowledge, such as complex medical terminology, clinical concepts, and reasoning patterns. For example, WSI-Agents (Lyu et al., 2025) leverages a "MLLM model library" comprising five pre-trained multimodal large language models (e.g., WSI-LLaVA (Liang et al., 2024) and Quilt-LLaVA (Seyfioglu et al., 2024)) specialized for Whole Slide Image (WSI) analysis. Other systems also rely on medically-informed base models, such as CardAIc-Agents (Zhang et al., 2025b) employs MedGemma (Sellergren et al., 2025) as the foundation for its multidisciplinary discussion tools.

**Model fine-tuning**: As the most intensive method, fine-tuning adapts models to highly specific medical tasks or datasets by further training them. This process adjusts the model's weights, enabling it to master specialized knowledge, adhere to specific clinical guidelines, or adopt a particular reporting style. It overcomes the generic nature of pre-trained models by instilling deep, task-specific expertise. For example, the agents in MMedAgent-RL (Xia et al., 2025) such as the "triage doctor" and "attending physician" are fine-tuned by using GRPO (Shao et al., 2024) (a kind of RL method) to impart domain knowledge and optimize collaborative policies and decision-making strategies based on feedback. MRGAgents (Wang et al., 2025a) fine-tunes base BioMedGPT (Luo et al., 2023) models for agents on dedicated disease-specific subsets in IU X-ray (Demner-Fushman et al., 2015) and MIMIC-CXR (Johnson et al., 2019) to improve medical report generation.

### 2.2.2 EXTERNALLY-ASSISTED

While agent-intrinsic methods enhance the inherent capabilities of LLMs, relying solely on the internal knowledge of these models presents significant challenges in the medical domain. LLMs are prone to factual hallucinations (Ji et al., 2023), lack the ability to process specialized data formats (e.g., genomic VCF files, ECG signals), and cannot execute deterministic computations required by many clinical protocols. To overcome these limitations, externally-assisted approaches equip agents

with the ability to call upon external tools, models, and knowledge bases. This paradigm allows the LLM to function as a central orchestrator or "brain", coordinating which external knowledge sources or tools to invoke.

This approach is exemplified by complex, hybrid systems that integrate multiple forms of external assistance. For instance, the DeepRare system (Zhao et al., 2025b) is designed for rare disease diagnosis by using an LLM as a central host that coordinates several specialized agents. These agents, in turn, invoke a variety of external medical tools and databases to perform evidence retrieval and diagnostic reasoning. Such systems illustrate a spectrum of external augmentation, which can be categorized by the increasing level of intelligence and complexity of the external resource, progressing from deterministic tools to specialized predictive models, and finally to dynamic knowledge retrieval systems.

**Traditional medicine tool utilization**: At the most fundamental level, agents employ established, often deterministic, medical tools as auxiliary supports. These tools include PubMed search engines, EHR retrieval systems, and clinical calculators. Their integration is critical for tasks requiring high fidelity, procedural accuracy, and the processing of structured or non-textual data. By offloading these functions, agents can ground their reasoning in reliable, standardized outputs, overcoming the non-deterministic and interpretive nature of LLMs. For exmaple, a genotype analysis agent in DeepRare (Zhao et al., 2025b) uses Exomiser (Smedley et al., 2015) (a specialized bioinformatics tool) to perform variant annotation and prioritization based on criteria like predicted pathogenicity, allele frequency, and genetic inheritance patterns. This process yields a precise list of candidate pathogenic genes, achieving a level of diagnostic accuracy in genomics that is unattainable for a generalist LLM. Similarly, the CardAIc-Agents framework (Zhang et al., 2025b) features a "CardiacExperts Agent" that utilizes NeuroKit2 (a Python toolbox) for processing raw ECG signals to obtain 12-leads ECG measurements.

**Domain-specific model calling**: A step beyond traditional tools, this approach involves agents integrating specialized AI models, such as radiology image classifiers or drug-disease interaction predictors. These models, often based on deep learning, provide sophisticated pattern recognition and predictive capabilities that complement the broad reasoning of an LLM. Calling these models allows the agent system to leverage deep, task-specific expertise learned from vast amounts of data, enabling more accurate analysis in domains like medical imaging or computational biology. For instance, DeepRare (Zhao et al., 2025b) calls upon PhenoBrain (Mao et al., 2025), a model that performs analysis on structured Human Phenotype Ontology (HPO) terms. Using classical machine learning and ontology matching, PhenoBrain rapidly generates an interpretable, probability-scored list of candidate diseases, enhancing the efficiency of the phenotype-driven diagnostic process by providing a quick and explainable differential diagnosis list for the LLM to reason upon. The CardiacExperts Agent in ardAIc-Agents (Zhang et al., 2025b) invokes multiple specialized models, including a fine-tuned multimodal cardiac diagnosis model considering lab data, ECG, and ultrasound, along with view classification and segmentation models for analyzing medical images.

**Medical knowledge-based RAG**: The most advanced form of external assistance involves Retrieval-Augmented Generation (RAG), where agents dynamically query knowledge bases and incorporate the retrieved information into their reasoning process at inference time. This method directly mitigates LLM hallucinations by grounding responses in factual, up-to-date information from structured (e.g., knowledge graphs) or unstructured (e.g., clinical guidelines, biomedical literature) sources. This ensures that the agent's outputs are not only contextually relevant but also transparent and traceable, fostering greater clinical trust. For example, DeepRare (Zhao et al., 2025b), employs a multi-faceted RAG strategy. Its knowledge retrieval agent implements an LLM-based RAG workflow to query medical knowledge bases like PubMed, Orphanet, and OMIM. The retrieved text is then summarized by a lightweight LLM to generate a transparent, traceable evidence chain that explicitly links diagnostic conclusions to source material. KERAP (Xie et al., 2025) uses a Retrieval Agent to extract and summarize entity relationships from a knowledge graph to inform diagnosis.

## 2.3 AGENT INTERACTION

Interaction mechanisms define how agents coordinate with each other to accomplish medical tasks. Effective interaction is critical for balancing efficiency and reliability in clinical contexts. We categorize existing approaches into two broad types: *hierarchical coordination*, where agents are orga-

nized in a layered structure (such as task delegators or aggregators that coordinate and integrate the work of other agents), and *peer collaboration*, where agents interact on a more equal footing.

### 2.3.1 HIERARCHICAL COORDINATION

**Agent recruitment**: In hierarchical settings, higher-level agents are responsible for instantiating or selecting subordinate agents with specific expertise to address a clinical scenario. This mechanism allows the system to dynamically assemble a team with the necessary domain knowledge. For example, an upper-level controller may recruit genetic and cardiovascular agents when facing a case involving both hereditary and symptomatic factors (Wang et al., 2025b). Similarly, in the MMedA-gent framework, a triage doctor first analyzes multimodal patient inputs to determine the appropriate specialty, and only the corresponding specialist agents (e.g., radiologists for X-ray images) are recruited, while others remain inactive (Xia et al., 2025). This targeted activation ensures efficient use of resources while still providing specialized reasoning for the clinical case.

**Task delegation**: Higher-level agents distribute subtasks to subordinate agents, thereby structuring the workflow into well-defined stages. This delegation enforces a clear division of labor: the main agent generates an initial diagnosis or hypothesis, then assigns targeted subtasks (e.g., evidence retrieval, calibration, or domain-specific reasoning) to domain experts. Subordinate agents return their findings, which are then aggregated by the higher-level agent (Zhao et al., 2025a). Such delegation improves efficiency by parallelizing subtasks while maintaining control at the supervisory level. For example, in a forensic investigation of a male fatality caused by aortic dissection after a car accident, the planner decomposes the case into subtasks such as analyzing rupture characteristics, linking trauma with pre-existing conditions (e.g., hypertension, coronary heart disease), ruling out poisoning, and differentiating rib fracture causes. These subtasks are then distributed to specialized solvers (e.g., autopsy analyzer, medical history integrator, toxicology interpreter, trauma classifier), each returning results that the planner synthesizes into the final conclusion (Shen et al., 2025).

**Joint discussion**: In addition to strict delegation, hierarchical systems may incorporate guided group deliberation, where higher-level agents act as moderators or evaluators of subordinate discussions. Usually, rather than participating as equals, subordinate agents can contribute candidate solutions, while the higher-level agent comments, provides feedback, and ensures alignment with the overarching diagnostic goal. This structure balances open exchange with centralized oversight, ensuring that junior agents' opinions are considered without compromising clinical reliability. For example, in a clinical setting, junior residents propose and critique preliminary diagnoses, while a senior doctor moderates the exchange, identifies cognitive biases, and steers the group toward a refined conclusion, supported by a recorder who consolidates outcomes Ke et al. (2024a).

**Information integration**: Finally, hierarchical coordination culminates in the synthesis of subordinate outputs into a unified decision or recommendation. The higher-level agent integrates intermediate results, weighing the evidence and resolving conflicts across subordinate reports. This step is crucial in clinical contexts, as it ensures that diverse sources of reasoning—such as genetic evidence, clinical manifestations, and patient history—are combined into a coherent and trustworthy medical conclusion (Chen et al., 2025a). By maintaining a supervisory "review–integrate–decide" cycle, hierarchical systems promote both interpretability and accountability. For example, in the MAM framework, the diagnostic process is decomposed into multiple specialized roles—including general practitioners, specialist teams, radiologists, medical assistants, and a chief physician—each embodied by an LLM-based agent. The chief physician coordinates the discussion, synthesizes opinions and retrieved evidence into interim reports, and the specialist group votes on whether to endorse these reports. Once consensus is reached, the chief physician consolidates the results into the final diagnosis (Zhou et al., 2025b).

### 2.3.2 PEER COLLABORATION

**Collaborative discussion**: Agents interact in a non-hierarchical manner without predefined workflows, exchanging ideas, sharing information, and jointly exploring solutions. Specifically, agents operate on an equal level, allowing for free exchange of ideas and information to refine the reasoning processes of each other and reduce hallucinations in clinical contexts. There are no designated leader and workflow, promoting a more democratic and inclusive discussion. A common form of collaborative discussion is peer collaboration within a multidisciplinary team (MDT) struc-

ture, where agents role-play as experts from various disciplines. For instance, Chen et al. (2025b) proposed SeM-Agents, which includes different doctor roles and auxiliary roles to facilitate collaborative discussions. In this system, the doctor agent team is organized based on the specific situation of the patient, enabling multi-round discussions among multidisciplinary doctors. Once a consensus is reached, a summary agent reviews and presents the results. Another example of collaborative discussion in the human-computer interaction field is presented by Li et al. (2025a). They propose a multi-agent system for answering medical questions and predicting diagnoses, where a human physician collaborates with medical expert agents from diverse backgrounds. Together, they debate and generate diagnostic results, assisting the human physician in making comprehensive decisions.

**Clinical task-driven flow**: Agent communication aligns with the stages of a real-world clinical workflow, with information being passed and updated according to task progression. This flow typically includes stages such as triage, consultation, and diagnosis. At each stage, agents share relevant data, insights, and findings, enhancing decision-making and promoting efficient patient management. This structured approach streamlines communication and helps maintain focus on clinical objectives, ultimately improving patient outcomes. For example, Xia et al. (2025) proposed MMedAgent-RL, which classifies agents into triage doctor, specialist doctor, and attending physician roles. The triage doctor agent performs initial departmental triage, routing patients to the appropriate department for diagnosis. Specialist doctor agents discuss the patient's symptoms and provide diagnostic advice, while the attending physician agent makes the final diagnosis based on the discussions. This entire process follows a standard clinical task flow: triage → specialist consultation → attending doctor diagnosis.

**Problem-solving cycles**: Agent communication follows the phases of a problem-solving workflow, which can be either sequential or iterative. This cycle typically includes stages such as planning, execution, evaluation, and refinement. During each phase, agents collaborate by sharing information and insights to address challenges encountered in the diagnostic process. This collaboration enables them to assess the effectiveness of their actions and make necessary adjustments. For example, ClinicalAgent (Yue et al., 2024) has a goal of predicting clinical trial outcomes. The framework decompose the task into three sequential sub-tasks: task decomposition, subproblem solving, and reasoning. ClinicalAgent introduces a planning agent to decompose the clinical trial task into several sub-tasks and then recruit enrollment agent, safety agent, and efficacy agent to solve the sub-tasks from different perspectives. Finally, it provides a reasoning agent to make final decisions of the clinical trial outcomes. Rather than following a standard clinical task flow, ClincialAgent follows a "divide-and-conquer" philosophy: it decomposes the whole process into three sub-stages and designs specific agents to solve each sub-task. Another example is HealthFlow (Zhu et al., 2025), which provides a self-evolving iterative problem-solving workflow. A meta-agent takes task input and generate actionable plans. The generated plans are executed by executor agents and the execution results are processed by the evaluator agent to obtain feedback. A reflection agent takes the feedbacks and provides experience to the meta agent to refine the actionable plans. The whole workflow is task-oriented and self-evolving.

## 3 DISCUSSION

In this section, we outline the main challenges and opportunities for advancing LLM-based multi-agent systems in medicine.

### 3.1 DESIGN AND EVALUATION OF AGENT PROFILES

A critical step in building LLM-based multi-agent systems is the design of agent profiles. In medical contexts, this often involves assigning agents specific backgrounds, such as different specialties (e.g., radiology, pathology) or junior versus senior roles, reflecting the hierarchical and collaborative nature of real-world clinical practice. Despite its importance, research on how to design, evaluate, and measure these roles remains limited. For instance, current approaches, including role-playing with LLMs or fine-tuning on domain-specific corpora, aim to enhance the knowledge representation and role-playing capabilities of LLMs, yet it is unclear how faithfully they capture the subtleties of medical reasoning, inter-disciplinary dynamics, and decision-making authority. Evaluation should extend beyond task accuracy to assess role fidelity—for example, whether junior agents exercise appropriate caution or senior agents provide credible oversight. Role authenticity also remains challenging:

agents may mimic the language of specialists without demonstrating genuine domain-consistent reasoning, producing superficially authoritative but potentially misleading outputs. Moreover, most systems assume static roles, whereas clinical responsibilities often shift dynamically depending on case complexity and team composition. Designing agent profiles that balance realism, adaptability, and reliability remains an open challenge.

## 3.2 Self-evolving Agentic Systems

Most existing multi-agent systems still rely on predefined architectures, static coordination strategies, and fixed agent-level configurations (e.g., knowledge bases, memory mechanisms, and reasoning pipelines), which inherently limit their ability to cope with change. The recent paradigm of self-evolving agents, however, highlights the possibility for agents to dynamically adapt and reorganize themselves. This capability is particularly important in medical contexts, where treatment decisions rarely remain static. As advances in medical technologies, the accumulation of clinical evidence, and the continuous revision of practice guidelines reshape therapeutic choices, agents must be able to autonomously update their knowledge bases, reasoning pathways, and coordination mechanisms. A small number of studies have begun to refine agent coordination and communication through reinforcement learning (Xia et al., 2025), yet much more work is needed to explore continual adaptation mechanisms, long-term knowledge integration, and autonomous strategy evolution. Additionally, considering the stringent safety requirements in medical settings, an equally important challenge lies in ensuring that such updates are transparent, verifiable, and interpretable. Unlike other domains where autonomous adaptation may be tolerated with minimal oversight, in medicine every adjustment to an agent's knowledge or reasoning pipeline can directly influence patient outcomes. Thus, it is crucial that the evolution of agentic systems be accompanied by mechanisms that not only record and justify how new knowledge is incorporated, but also allow clinicians to audit, validate, and, when necessary, override these updates. Beyond interpretability at the decision level, transparency must also extend to the processes of coordination and knowledge integration across agents, so that the entire multi-agent system remains trustworthy and accountable.

## 3.3 Human Intervention

Much of the current research on LLM-based multi-agent systems in medical tasks has focused on fully automated approaches. However, such methods are inherently limited: they are vulnerable to hallucinations, cannot fully capture tacit clinical expertise, and risk reducing clinicians to passive validators. In high-stakes domains such as medicine, where safety and domain expertise are significant, human-in-the-loop mechanisms remain indispensable for ensuring clinical reliability, accountability, and trustworthiness (Xu et al., 2025). A key limitation of current designs is that human involvement is often reduced to a final validation step, which treats clinicians as passive overseers rather than active collaborators. This raises several open questions: at what stages should human expertise be integrated; how much control should clinicians retain; and how can systems balance efficiency with safety? Without careful design, excessive reliance on physicians could increase workload, while too little involvement may erode trust. Future LLM-based multi-agent systems should therefore embrace human–AI collaboration as a design principle. Instead of restricting clinicians to a final validation step, systems should support interactive modes where experts can dynamically steer agent discussions, inject expertise knowledge, or arbitrate conflicts between divergent agent outputs. By advancing toward this participatory paradigm, multi-agent systems can move beyond static decision-support and evolve into partners in clinical reasoning.

## 3.4 Multimodal Integration

Clinical decision-making rarely depends on a single source of information. Instead, it synthesizes evidence from multiple modalities, such as imaging, genomic sequences, laboratory results, and electronic health records (EHRs). Some recent multi-agent medical systems have begun to incorporate multimodal inputs (Xia et al., 2025; Zhang et al., 2025b). However, these systems handle these modalities as isolated inputs, each contributing to decision-making, without considering how they interact with one another. In practice, the interplay between modalities is far more complex. Multimodal information may exhibit different relationships: dominance (one modality outweighing others), complementarity (different modalities reinforcing each other), or conflict (modalities

providing contradictory evidence) (Wang et al., 2021). Thus, future research should move beyond simple aggregation and develop mechanisms to explicitly identify, reconcile, and leverage these relationships to support more robust and trustworthy multi-agent reasoning in clinical contexts.

## 3.5 INTERACTION PATTERNS

The way agents interact is central to the reliability of medical multi-agent systems. Existing work has largely explored two modes. Hierarchical coordination offers efficiency and accountability by assigning supervisory agents to recruit, delegate, and integrate, but it risks error propagation if oversight is flawed. Peer collaboration, in contrast, promotes balanced participation and robustness through mutual critique, yet often struggles with conflict resolution. Between these two extremes, a widely adopted form is the multidisciplinary team (MDT), which mirrors real-world clinical consultations. MDTs can be understood as a structured instantiation of peer collaboration with elements of hierarchy: cases are presented, specialists contribute in turn, conflicting views are deliberated, and a chair physician synthesizes the outcome. However, current implementations rarely capture these procedural norms, often reducing MDT to unconstrained dialogue (Wang et al., 2025c; Kim et al., 2024; Zhang et al., 2025a). The key challenge, then, is not only to scale MDT-style collaboration but also to translate its well-established practices into agent workflows. Future work should investigate how to (1) design structured discussion protocols that mirror clinical MDT flow, (2) develop systematic mechanisms for resolving conflicting specialist outputs (e.g., weighted voting, confidence calibration, or arbitration), and (3) combine supervisory oversight with structured peer exchange so that accountability is preserved while ensuring that diverse expert reasoning is fully represented.

## 3.6 MEDICAL SCENARIOS

Multi-agent systems have demonstrated growing application potential across a wide range of medical scenarios. Among these, diagnosis is the most extensively studied, ranging from general disease (Kim et al., 2024; Wang et al., 2025c) to domain-specific settings such as glaucoma detection (Liu et al., 2025), cardiology (Zhang et al., 2025a), and rare disease diagnosis (Chen et al., 2024). Beyond diagnosis, applications have also emerged in outpatient reception and triage (Lu et al., 2024; Bao et al., 2024), treatment planning and optimization (Chen et al., 2024; Yue et al., 2024; Chen et al., 2025f), and clinical decision-making (Ke et al., 2024a; Liu et al., 2024b). The objectives of these studies are diverse: some aim to reduce cognitive bias (Ke et al., 2024b; Li et al., 2025a), others focus on mitigating hallucinations (Low et al., 2025a), improving explainability (Liu et al., 2024b; Hong et al., 2024), or providing more personalized medical services (Bao et al., 2024). Looking ahead, the advancement of multi-agent systems should expand to a broader range of medical tasks and scenarios that go beyond what clinicians can achieve. For instance, tasks such as efficacy and prognosis prediction in cancer immunotherapy, or the assessment of metastasis and recurrence risks are often difficult to address through clinical experience alone due to the high heterogeneity of tumors. These challenges are also suited to agent-based decision-making supported by models trained on large-scale medical datasets. Moreover, health education plays a critical role in enhancing public understanding of diseases and promoting early interventions. Multi-agent systems, equipped with role modeling and language style adaptation capabilities, can generate medical content that is both accessible and personalized based on the target audience. Compared to human physicians, agents are more effective at translating specialized medical content into comprehensible, public-facing language, thereby improving the dissemination efficiency of health information.

## 4 CONCLUSION

This survey systematically reviews the development of LLM-based multi-agent systems for medical problem-solving. We develop a medical-specific taxonomy along three dimensions: team composition, medical knowledge enhancement, and agent interactions, from our analysis of 50 papers. Despite recent progress, challenges remain in designing domain-specific agents and interactions. Future research should address these gaps, such as incorporating human–AI collaboration to ensure that human experts and multi-agent systems jointly address complex clinical tasks. We hope this work lays the groundwork for advancing reliable, impactful, and practically usable multi-agent systems in medicine.

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

# A APPENDIX

## A.1 TAXONOMY

Table 1 summarizes our taxonomy of medical multi-agent systems for problem-solving across three dimensions: team composition, knowledge enhancement, and agent interaction.

Figure 1 presents the taxonomy-based coding of the surveyed papers in detail.

## A.2 ACKNOWLEDGMENTS OF THE USE OF LLM

In this paper, we used ChatGPT to check grammar and improve wording. It did not change the original meaning of the text or introduce any new references or knowledge.

## A.3 ETHICS STATEMENT

Our survey does not involve human subjects, personal data, or sensitive information, and therefore does not raise direct ethical concerns. All analyzed papers are publicly available under appropriate licenses, and no private or identifiable information is included. We have carefully considered potential risks of bias, fairness, and misuse, and conclude that our work adheres to the ICLR Code of Ethics.

## A.4 REPRODUCIBILITY STATEMENT

We have taken concrete measures to ensure the reproducibility of our results. In particular, we provide the detailed encodings of all surveyed papers in Figure 1, along with comprehensive descriptions of the taxonomy construction process in the main text (Section 2). These materials collectively allow researchers to verify and extend our findings.

Table 1: Taxonomy of LLM-based Multi-agent Systems in Medicine

| Category | Subcategory | Sub-subcategory | Related Work |
|---|---|---|---|
| Team Composition | | Clinical task allocation | Iapascurta et al. (2025), Bao et al. (2024), Yue et al. (2024), Xia et al. (2025), Zhao et al. (2025b), Wang et al. (2025a), Jia et al. (2025), Zhang et al. (2025b), Tang et al. (2023), Yan et al. (2024), Low et al. (2025a), Chen et al. (2025f), Chen et al. (2025e), Ghezloo et al. (2025) |
| | | Specialization-oriented assignment | Kim et al. (2024), Zhou et al. (2025b), Li et al. (2025b), Chen et al. (2025b), Xia et al. (2025), Zhao et al. (2025b), Mishra et al. (2025), Wang et al. (2025a), Zhang et al. (2025b), Li et al. (2025a), Tang et al. (2023), Chen et al. (2024), Bao et al. (2025), Yang et al. (2025), Liu et al. (2025), Lyu et al. (2025), Zhao et al. (2025c), Solovev et al. (2024), Zuo et al. (2025) |
| | | Process-oriented allocation | Wang et al. (2025c), Zhang et al. (2025a), Hong et al. (2024), Liu et al. (2024a), Ke et al. (2024a), Xu et al. (2025), Low et al. (2025b), Zhao et al. (2025a), Mishra et al. (2025), Li et al. (2025a), Wu et al. (2024), Liang et al. (2025), Zhou et al. (2024), Zhou et al. (2025a), Xie et al. (2025), Liu et al. (2024b), Lyu et al. (2025), Shi et al. (2024), Chen et al. (2025c), Mahajan & Ji (2025), Solovev et al. (2024), Chen et al. (2025a), Zhu et al. (2025), Ghezloo et al. (2025) |
| | | Expertise-level assignment | Ke et al. (2024b), Wang et al. (2025e), Low et al. (2025b), Bao et al. (2025), Chen et al. (2025c) |
| | | Automatic assignment | Wang et al. (2025c), Chen et al. (2025b), Zhao et al. (2025a), Mishra et al. (2025), Li et al. (2025a), Tang et al. (2023), Yan et al. (2024), Bao et al. (2025), Yang et al. (2025), Chen et al. (2024), Lyu et al. (2025), Chen et al. (2025c), Zhu et al. (2025) |
| Medical Knowledge | Agent-intrinsic | Role-play prompting | Kim et al. (2024), Wang et al. (2025c), Zhou et al. (2025b), Ke et al. (2024a), Ke et al. (2024b), Lu et al. (2024), Li et al. (2025b), Chen et al. (2025b), Xia et al. (2025), Mishra et al. (2025), Li et al. (2025a), Tang et al. (2023), Yan et al. (2024), Chen et al. (2024), Liu et al. (2025), Chen et al. (2025c), Zhao et al. (2025c) |
| | | Pre-trained model utilization | Iapascurta et al. (2025), Liu et al. (2024a), Wang et al. (2025a), Zhang et al. (2025b), Lyu et al. (2025), Mahajan & Ji (2025), Ghezloo et al. (2025) |
| | | Model fine-tuning | Bao et al. (2024), Zhou et al. (2025b), Wang et al. (2025e), Li et al. (2025b), Xia et al. (2025), Wang et al. (2025a), Jia et al. (2025), Zhang et al. (2025b), Liang et al. (2025), Chen et al. (2025a), Ghezloo et al. (2025) |
| | Externally-assisted | Traditional medicine tool utilization | Zhang et al. (2025a), Hong et al. (2024), Zhou et al. (2025a), Liu et al. (2025), Chen et al. (2024) |
| | | Domain-specific model calling | Wang et al. (2025e), Yue et al. (2024), Zhao et al. (2025b), Zhang et al. (2025b), Liang et al. (2025), Zhou et al. (2025a), Chen et al. (2024), Lyu et al. (2025), Mahajan & Ji (2025), Solovev et al. (2024) |
| | | Medical knowledge-based RAG | Iapascurta et al. (2025), Bao et al. (2024), Liu et al. (2024a), Wang et al. (2025e), Lu et al. (2024), Yue et al. (2024), Zhao et al. (2025b), Zhao et al. (2025a), Jia et al. (2025), Li et al. (2025a), Low et al. (2025a), Chen et al. (2025f), Liang et al. (2025), Zhou et al. (2024), Zhou et al. (2025a), Xie et al. (2025), Liu et al. (2024b), Lyu et al. (2025), Shi et al. (2024), Zhao et al. (2025c), Mahajan & Ji (2025), Solovev et al. (2024), Chen et al. (2025a), Zhu et al. (2025), Zuo et al. (2025) |
| Agent Interaction | Hierarchical Coordination | Agent recruitment | Chen et al. (2025b), Xia et al. (2025), Zhao et al. (2025b), Low et al. (2025b), Zhao et al. (2025a), Mishra et al. (2025), Chen et al. (2024), Chen et al. (2025e), Chen et al. (2025c), Chen et al. (2025a), Zuo et al. (2025) |
| | | Task delegation | Yue et al. (2024), Zhao et al. (2025b), Zhao et al. (2025a), Zhang et al. (2025b), Chen et al. (2025c) |
| | | Joint discussion | Ke et al. (2024a), Wang et al. (2025e), Lu et al. (2024), Mishra et al. (2025), Zhang et al. (2025b), Chen et al. (2025c), Zhu et al. (2025) |
| | | Information integration | Kim et al. (2024), Zhou et al. (2025b), Hong et al. (2024), Liu et al. (2024a), Ke et al. (2024b), Wang et al. (2025e), Lu et al. (2024), Li et al. (2025b), Chen et al. (2025b), Yue et al. (2024), Xia et al. (2025), Zhao et al. (2025b), Zhao et al. (2025a), Mishra et al. (2025), Zhang et al. (2025b), Bao et al. (2025), Yang et al. (2025), Liu et al. (2025), Chen et al. (2024), Liu et al. (2024b), Lyu et al. (2025), Shi et al. (2024), Chen et al. (2025c), Mahajan & Ji (2025), Solovev et al. (2024), Chen et al. (2025a), Zhu et al. (2025), Ghezloo et al. (2025), Zuo et al. (2025) |
| | Peer Collaboration | Collaborative Discussion | Kim et al. (2024), Wang et al. (2025c), Zhou et al. (2025b), Zhang et al. (2025a), Ke et al. (2024a), Wang et al. (2025e), Chen et al. (2025b), Li et al. (2025a), Tang et al. (2023), Yan et al. (2024), Chen et al. (2024), Low et al. (2025a), Bao et al. (2025), Lyu et al. (2025), Chen et al. (2025c), Zhao et al. (2025c), Ghezloo et al. (2025) |
| | | Clinical task-driven flow | Iapascurta et al. (2025), Bao et al. (2024), Ke et al. (2024b), Wang et al. (2025e), Xia et al. (2025), Zhao et al. (2025a), Mishra et al. (2025), Jia et al. (2025), Zhang et al. (2025b), Tang et al. (2023), Yan et al. (2024), Chen et al. (2024), Chen et al. (2025f), Chen et al. (2025e), Ghezloo et al. (2025) |
| | | Problem-solving cycles | Lu et al. (2024), Li et al. (2025b), Xu et al. (2025), Zhao et al. (2025b), Low et al. (2025b), Zhao et al. (2025a), Mishra et al. (2025), Wang et al. (2025a), Zhang et al. (2025b), Wu et al. (2024), Low et al. (2025a), Liang et al. (2025), Zhou et al. (2024), Zhou et al. (2025a), Xie et al. (2025), Liu et al. (2024b), Lyu et al. (2025), Shi et al. (2024), Chen et al. (2025c), Mahajan & Ji (2025), Solovev et al. (2024), Chen et al. (2025a), Zhu et al. (2025), Zuo et al. (2025) |

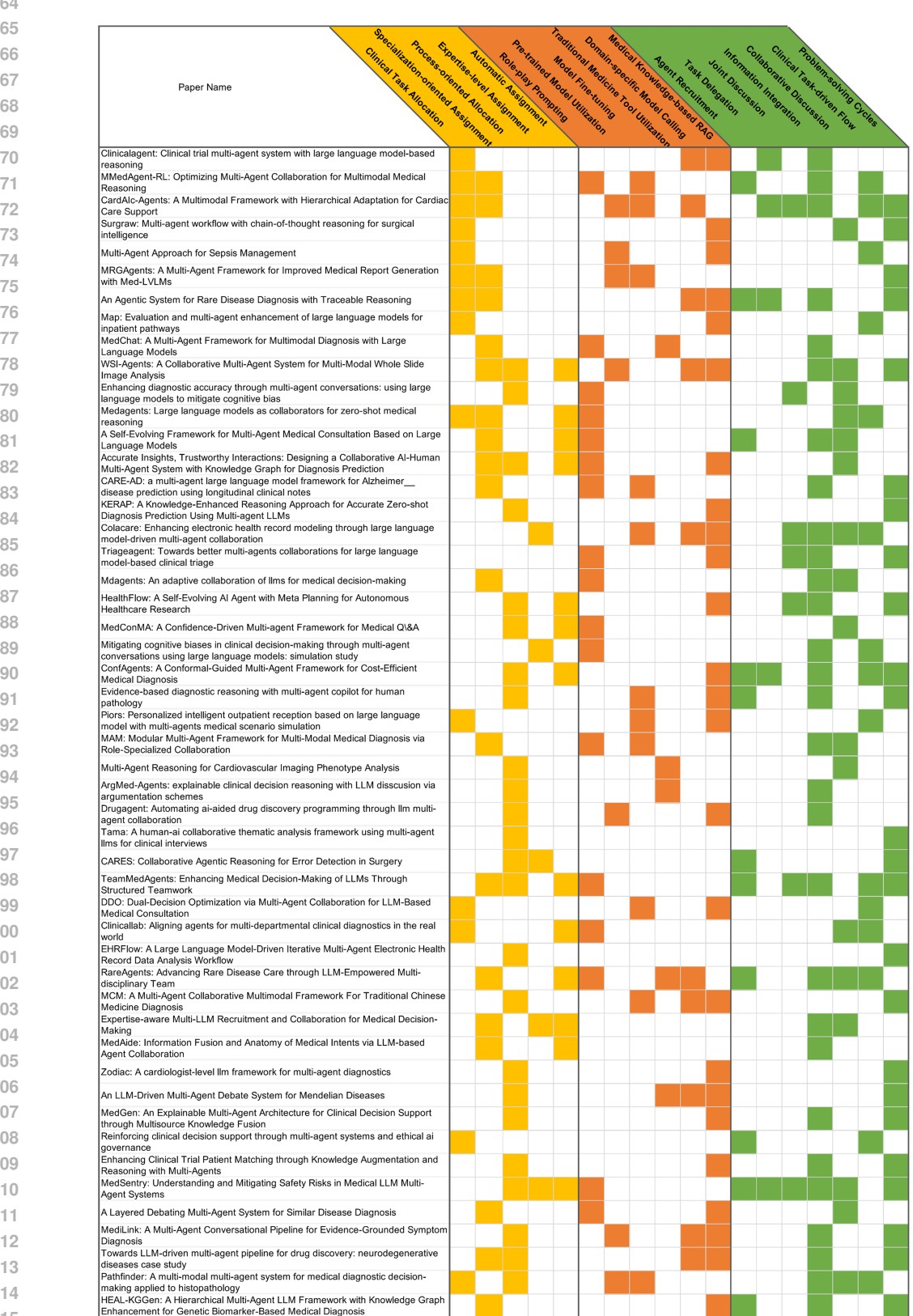

Figure 1: The coding of each paper under the proposed taxonomy.

