# OpenReview forum: "A Survey of LLM-based Multi-agent Systems in Medicine"
_ICLR.cc/2026/Conference — ICLR 2026 Conference Withdrawn Submission_

### Official Review · Reviewer_oeyv · 2025-10-31

**Soundness:** 2
**Presentation:** 2
**Contribution:** 2
**Rating:** 2
**Confidence:** 5

**Summary:**

This paper surveys 50 recent works on LLM-based multi-agent systems in medicine. The authors propose a medical-specific taxonomy structured around three key design dimensions: team composition, medical knowledge augmentation, and agent interaction.

**Strengths:**

The paper's primary strength is its development of a structured, medical-specific taxonomy that addresses the shortcomings of existing domain-agnostic surveys.

**Weaknesses:**

My major comments are:

1. The criteria for excluding papers are not fully justified. For instance, the authors excluded works on "purely biological or genomic research". This is a questionable omission, as LLM-based agents are increasingly used in areas like drug discovery and genetic analysis, which are highly relevant to medical problem-solving.
2. The survey explicitly excludes multi-agent systems for hospital simulation, noting that other surveys focus on this topic. However, these systems, which model hospital operations or clinical workflows, represent a significant application of agend-based framework in medicine. The survey should at least briefly summarized this area or provided a stronger justification for its complete exclusion.
3. The survey largely avoids a critical or comparative evaluation of the approaches it identifies. The paper does not systematically discuss which design patterns (e.g., which team composition or interaction style) are more effective, reliable, or safe for specific medical tasks. The discussion is very superficial. It critiques current MDT models as unconstrained dialogue but only offers general recommendations like designing structured discussion protocols without a deeper analysis of how existing systems fail or what specific mechanisms could resolve this.

**Questions:**

Please refer to my comments above.

---

### Official Review · Reviewer_95aj · 2025-11-02

**Soundness:** 2
**Presentation:** 1
**Contribution:** 1
**Rating:** 2
**Confidence:** 5

**Summary:**

This paper surveys the burgeoning literature on Large Language Model (LLM)-based multi-agent systems for medical problem-solving. It introduces a design-oriented taxonomy spanning three dimensions: team composition, medical knowledge augmentation, and agent interaction, using a corpus of 50 curated and coded papers. The survey offers an analytic framework, discusses prominent paradigms, and exposes open challenges.

**Strengths:**

1. The discussion sections identify key open questions unique to medical applications, such as agent profile fidelity, self-evolution, multimodal integration, and the subtleties of human-AI collaboration.
2. Figure 1 is a useful one-glance visual of the surveyed papers' coverage across taxonomy axes.

**Weaknesses:**

1. While the taxonomy is systematic, sections could do more analytical heavy lifting to distinguish why certain paradigms (e.g., specialization vs. process-based agent allocation) are preferable or inferior in specific medical contexts. The discussion remains too high-level in places and often defers to representative examples.

2. Although the paper states that two reviewers independently coded and then reached consensus, the criteria and process for merging or splitting taxonomy categories (e.g., how team composition subtypes were defined and unified) are not rigorous.

3. While the discussion touches on hallucination, error propagation, and trust, the paper overlooks a deeper, focused analysis of empirical and technical failure modes emerging in practice.

4. Despite the extensive bibliography, several highly relevant recent surveys and foundational works on LLM-based multi-agent systems remain unreferenced, such as:
- A Survey of LLM-based Agents in Medicine: How far are we from Baymax?
- LLM-Based Human-Agent Collaboration and Interaction Systems: A Survey
 - A Survey on Large Language Model Based Autonomous Agents

5. The survey's treatment of algorithmic and methodological patterns is superficial. It lacks systematic analysis of mathematical formulations, such as agent-interaction protocols, convergence or generalization theorems, and standardized loss functions. Beyond empirical categorizations, the paper should also classify prevailing theoretical design choices and discuss their implications (e.g., objective functions, guarantees). Without this, readers cannot effectively compare methods at a technical level.

6. Overall, the work falls considerably short of the rigor and depth typically expected at the ICLR level.

**Questions:**

see Weaknesses

---

### Official Review · Reviewer_Wg4K · 2025-11-05

**Soundness:** 2
**Presentation:** 1
**Contribution:** 2
**Rating:** 0
**Confidence:** 4

**Summary:**

This paper presents a survey of LLM-driven multi-agent systems in medicine. Its central contribution is a proposed three-dimensional, medical-specific taxonomy intended to organize design patterns in this domain. The authors posit that existing, domain-agnostic taxonomies are insufficient to capture the unique characteristics of medical applications.

While the paper is clearly structured and follows a transparent literature search methodology, its core contribution unfortunately lacks the necessary novelty and insight required for ICLR. The proposed taxonomy fails to substantiate its medical-specific claim, and the paper as a whole reads more as a well-organized literature collection than a critical synthesis. This fundamental weakness, compounded by significant presentation issues, renders the paper's contribution insufficient.

**Strengths:**

1. Timely Literature Collection: The paper gathers a comprehensive set of recent works (50 papers) in the emerging and important field of medical LLM multi-agent systems. The literature search methodology is clearly articulated.

2. Identifies Future Directions: The discussion section (Section 3) clearly outlines several relevant and important future research challenges (e.g., human-AI collaboration, self-evolution, multimodal integration), which serves as a useful reference for researchers.

3. Structural Attempt: The paper attempts to provide a classification framework, offering a basic structure for organizing the reviewed literature.

**Weaknesses:**

The paper suffers from several major weaknesses, the most significant being the lack of a novel.

## The Core Contribution is Not Medical-Specific and Lacks Novelty.

The paper's primary claim is that existing domain-agnostic taxonomies are insufficient. However, the proposed three-dimensional taxonomy fails to substantiate this claim. Instead, it appears to be a direct recapitulation of standard, domain-agnostic concepts from Multi-Agent Systems (MAS) and general LLM engineering, which are then populated with medical examples. This renders the framework a tool for summarization, not the critical synthesis expected of a top-tier survey.

- **Team Composition (Sec 2.1)**: The categories (e.g., Clinical task allocation, Specialization-oriented assignment, Process-oriented allocation) are standard organizational and workflow patterns in MAS design, equally applicable to any domain (e.g., finance, logistics, gaming).

- **Medical Knowledge Augmentation (Sec 2.2)**: The Agent-Intrinsic vs. Externally-Assisted split is a generic classification for all RAG/LLM augmentation methods. The paper fails to synthesize why these methods are fundamentally different or present unique challenges in medicine (e.g., specific demands for evidence traceability, real-time constraints, or unique data modalities) compared to their use in other fields.

- **Agent Interaction (Sec 2.3)**: Hierarchical Coordination and Peer Collaboration are foundational, classic concepts in MAS. The paper simply slots medical examples (like MDT discussions) into these pre-existing buckets without offering a new perspective or identifying emergent, medicine-specific interaction patterns.

Ultimately, the reader gains no new insight into the unique design patterns or challenges specific to medical MAS. The paper fails to answer the very question it poses: *why are general taxonomies insufficient?*

## Poor Presentation and Lack of Synthesis.

The presentation style severely hinders the paper's readability and underscores the lack of synthesis.

- **Repetitive Listing**: The core taxonomy sections (2.1-2.3) are written in a highly repetitive, list-like style (e.g., A did X, B did Y, C did Z) that merely restates paper abstracts. There is almost no critical commentary, comparison, or synthesis connecting these individual works.

- **Lack of Visual Aids**: The main body of the paper lacks any summary figures or tables. The only two visualizations are relegated to the appendix. For a survey paper, which aims to clarify and structure a field, this is a major omission. A visual representation of the taxonomy or a comparative table mapping the 50 papers to the framework's dimensions is essential for providing information gain.

**Questions:**

1. The central claim of the paper is that this taxonomy is medical-specific and that existing general taxonomies are insufficient. Could you please provide a clear justification for this? Specifically, how do the proposed categories fundamentally differ from standard, domain-agnostic MAS taxonomies? What specific aspects of medical MAS do existing frameworks fail to capture that this one successfully does?

2. Given that this is a survey paper, the complete lack of figures or tables in the main body is a critical flaw for readability. Could you elaborate on the decision to place all visualizations in the appendix? A comparative table or diagram is essential for a reader to understand the relationships between the 50 surveyed papers and the proposed framework.

---

### Official Review · Reviewer_T2sE · 2025-11-05

**Soundness:** 3
**Presentation:** 2
**Contribution:** 1
**Rating:** 2
**Confidence:** 3

**Summary:**

This paper presents a survey of LLM-based multi-agent systems in the medical domain. It analyzes 50 recent papers to propose a novel, medical-specific taxonomy organized along three dimensions: team composition, medical knowledge augmentation, and agent interaction. Based on this taxonomy, the paper discusses existing challenges and identifies several promising future research directions, including human-AI collaboration, self-evolving systems, and multimodal integration, aiming to provide a structured overview to guide future work in this rapidly developing field.

**Strengths:**

- The survey addresses a timely and highly significant topic at the intersection of large language models, multi-agent systems, and medicine.
- It provides a useful snapshot of a rapidly evolving field by synthesizing a considerable number of very recent publications.
- The high-level dimensions of the taxonomy—team composition, knowledge augmentation, and interaction—offer a logical starting point for structuring the domain.
- The discussion section successfully identifies several important and relevant challenges that are critical for the future development of medical multi-agent systems.

**Weaknesses:**

- The proposed taxonomy for "Team Composition" contains categories that are not mutually exclusive and operate at different levels of abstraction.
- The subcategories within "Agent Interaction" incorrectly mix interaction patterns with workflow descriptions, creating a confusing and structurally unsound classification.
- The paper selection process appears limited, relying on a single search engine and snowballing, which may introduce selection bias and miss relevant literature.
- A substantial portion of the surveyed works are non-peer-reviewed preprints, which undermines the scholarly foundation and authority expected from a survey paper.
- The analysis lacks a deep critical perspective, functioning more as a descriptive catalog of recent work rather than an evaluation of their comparative strengths and weaknesses.
- The discussion of future challenges feels disconnected from the survey's taxonomic findings, presenting generic AI research topics without linking them back to the specific patterns identified.
- There are apparent misclassifications in the examples provided, such as describing systems with clear hierarchical roles like a triage doctor under the umbrella of peer collaboration.

**Questions:**

- Could you clarify the principle for categorization within the "Team Composition" dimension when a single system exhibits multiple characteristics, such as using automatic assignment to select agents based on both their specialization and expertise level?
- Your "Agent Interaction" taxonomy places "Clinical task-driven flow" under "Peer Collaboration," but real-world clinical workflows often involve hierarchical steps like triage and referral. Can you elaborate on the justification for this classification?
- How did you address the potential for unverified claims and methodological weaknesses in the large number of non-peer-reviewed preprints included in your analysis, and how might this affect the validity of your proposed taxonomy?

---

### Official Review · Reviewer_jDVj · 2025-11-11

**Soundness:** 2
**Presentation:** 2
**Contribution:** 2
**Rating:** 4
**Confidence:** 4

**Summary:**

This survey reviews 50 papers on LLM-based multi-agent systems for medical problem-solving. It proposes a taxonomy across three dimensions: team composition, medical knowledge augmentation, and agent interaction. This taxonomy complements the current use of more general multi-agent taxonomies, which does not adequately capture medical-specific patterns such as multidisciplinary team collaboration and clinical workflows. The paper provides a good framework for characterizing design patterns in medical multi-agent systems, as well as proposes future directions including human-AI collaboration, self-evolving systems, and multimodal integration.

**Strengths:**

- The paper specifically focuses on multi-agent systems and interactions in medicine, which complements existing survey papers in the field and provides additional insights on the design of multi-agent systems. It clearly compares and contradicts its contributions with existing papers.
- The paper provides design-focused takeaways, such as "how to compose teams" and "how to enable interaction". This could be useful for system builders.

**Weaknesses:**

1. Lack of visualization in the paper.
- The main paper does not have any figures or tables, which I think makes the paper a bit hard to read. It would be very helpful to have a flow diagram explaining your methodology of selecting and categorizing papers, as well as a figure showing the three dimensions and the subcategories that fall underneath each.
- Is there a way to have a condensed version of Table 1 or Figure 1 in the main paper?

2. Limited analytical depth.
- The paper mainly *describes* existing work and explain why they fall under the proposed taxonomy, but there is little critical assessment of the papers and meta-level discussions on the limitations of these works. There is also little comparative analysis of the papers, such as showing which design combinations are effective.

3. The methodology of designing the taxonomy is unclear.
- The paper treats the proposed taxonomy as a given, without explaining how they came up with the three dimensions and the criteria used.

**Questions:**

1. How does "clinical task-driven flow" differ from "problem-solving cycles"?

2. How did you come up with the three dimensions (team composition, medical knowledge enhancement, agent interaction)? (related to weakness 2)

---

### Note · Authors · 2025-11-12

I have read and agree with the venue's withdrawal policy on behalf of myself and my co-authors.